# FEATURE-MAP-LEVEL
# ONLINE ADVERSARIAL KNOWLEDGE DISTILLATION

## ABSTRACT

Feature maps contain rich information about image intensity and spatial correlation. However, previous online knowledge distillation methods only utilize the class probabilities. Thus in this paper, we propose an online knowledge distillation method that transfers not only the knowledge of the class probabilities but also that of the feature map using the adversarial training framework. We train multiple networks simultaneously by employing discriminators to distinguish the feature map distributions of different networks. Each network has its corresponding discriminator which discriminates the feature map from its own as fake while classifying that of the other network as real. By training a network to fool the corresponding discriminator, it can learn the other network's feature map distribution. Discriminators and networks are trained concurrently in a minimax two-player game. Also, we propose a novel cyclic learning scheme for training more than two networks together. We have applied our method to various network architectures on the classification task and discovered a significant improvement of performance especially in the case of training a pair of a small network and a large one.

## 1 INTRODUCTION

With the advent of Alexnet (Krizhevsky et al., 2012), deep convolution neural networks have achieved remarkable success in a variety of computer vision tasks. However, high-performance of deep neural network is often gained by increasing the depth or the width of a network. Deep and wide networks cost a large number of computation as well as memory storage which is not suitable for a resource-limited environment such as mobile or embedded systems. To overcome this issue, many researches have been conducted to develop smaller but accurate neural networks. Some of the well-known methods in this line of research are *parameter quantization or binarization* (Rastegari et al., 2016), *pruning* (Li et al., 2016) and *knowledge distillation* (KD) (Hinton et al., 2015).

KD has been an active area of research as a solution to improve the performance of a light-weight network by transferring the knowledge of a large pre-trained network (or an ensemble of small networks) as a teacher network. KD sets the teacher network's class probabilities as a target which a small student network tries to mimic. By aligning the student's predictions to those of the teacher, the student can improve its performance. Recently, some studies have shown that rather than using a pre-trained teacher, simultaneously training networks to learn from each other in a peer-teaching manner is also possible. This approach is called online distillation. *Deep mutual learning* (DML) (Zhang et al., 2018) and *on-the-fly native ensemble* (ONE) (Lan et al., 2018) are the representative online distillation methods that show appealing results in the image classification tasks. Conventional distillation method requires pre-training a powerful teacher network and performs an one-way transfer to a relatively small and untrained student network. On the other hand, in online mutual distillation, there is no specific teacher-student role. All the student networks learn simultaneously by teaching each other from the start of training. It trains with the conventional cross-entropy loss from the ground truth label along with the mimicry loss to learn from its peers. Networks trained in such an online distillation way achieve results superior not only to the networks trained with the cross-entropy loss alone but also to those trained in conventional offline distillation manner from a pre-trained teacher network.

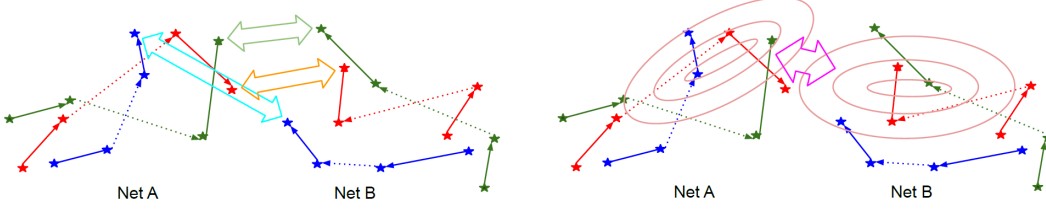

Net A                    Net B                              Net A                    Net B

(a)    Direct Feature Map Alignment        (b)    Adversarial Feature Map Distillation

Figure 1: **The concept of** *online Adversarial Feature map Distillation* **(AFD)** Each point represents a feature map for the corresponding input denoted by different colors. The thin line arrow indicates the evolvement of feature map data points as iteration goes on and the broader arrow indicates the way each method compares the feature maps from different networks. (a) In direct feature map alignment, networks are trained such that the distance between each pair of points with the same color is minimized. (b) In AFD, the discriminators contain information on feature map distributions and thus the networks are trained such that the distributions match. (best viewed in color)

However, aforementioned online distillation methods make use of only the logit information. While the logit contains the probabilistic information over classes, the feature map, the output of convolution layer, has more meaningful and abundant feature information on image intensity and spatial correlation. In offline distillation which utilizes a pre-trained model as a teacher network, many methods such as *FitNet* (Romero et al., 2014), *attention transfer* (AT) (Zagoruyko & Komodakis, 2016a) and *factor transfer* (FT) (Kim et al., 2018) make use of this intermediate feature representation as a target to learn for the student network, but in online distillation, to the best of our knowledge, no feature map-based knowledge distillation method has been proposed.

This is due to some challenges. Unlike the offline methods that have a clear target to mimic, there is no static target to follow in an online method. At every training iteration, the feature maps of the co-trained network change, thus in online feature map-level distillation, the problem turns into mimicking the moving target properly. While each node of the logit is confined to represent its assigned class probability which does not change drastically over iterations, at the feature map-level, much more flexibility comes into play, which makes the problem more challenging. Therefore, the direct aligning method such as using L1 or L2 distance is not suitable for online mutual feature map distillation because it updates the network parameters to generate a feature map that tries to mimic the current output feature map of the other network. In other words, the direct alignment method only tries to minimize the distance between the two feature map points (one for each network), hence it ignores the distributional difference between the two feature maps (Fig. 1(a)).

To alleviate this problem, in this paper, we propose a novel online distillation method that transfers the knowledge of feature maps adversarially as well as a cyclic learning framework for training more than two networks simultaneously. Unlike the direct aligning method, our adversarial distillation method enables a network to learn the overall feature map distribution of the co-trained network (Fig. 1(b)). Since the discriminator is trained to distinguish the difference between the networks' feature map distributions (containing the history of feature maps for different input images) at every training iteration, by fooling the discriminator, the network learns the co-trained network's changing feature map distribution. Exchanging the knowledge of feature map distribution facilitates the networks to converge to a better feature map manifold that generalizes better and yields more accurate results.

Our method consists of two major losses: 1) logit-based loss and 2) feature map-based loss. Logit-based loss is defined by two different loss terms which are conventional cross-entropy (CE) loss and the mutual distillation loss using the Kullback-Leibler divergence (KLD). Our newly proposed feature map-based loss is to distill the feature map indirectly via discriminators. We use the feature map from the last convolution layer since deeper convolution layer generates more meaningful features with a high-level abstraction (Kim et al., 2018). The adversarial training scheme of *generative adversarial networks* (GAN) (Goodfellow et al., 2014) is utilized to transfer the knowledge at feature map-level.

The contributions of this paper can be summarized as follows: 1) we propose an online knowledge distillation method that utilizes not only the logit but also the feature map from the convolution layer. 2) Our method transfers the knowledge of feature maps not by directly aligning them using

the distance loss but by learning their distributions using the adversarial training via discriminators. 3) We propose a novel cyclic learning scheme for training more than two networks simultaneously.

## 2 RELATED WORK

The idea of *model compression* by transferring the knowledge of a high performing model to a smaller model was originally proposed by Buciluǎ et al. (2006). Then in recent years, this research area got invigorated due to the work of *knowledge distillation* (KD) by Hinton et al. (2015). The main contribution of KD is to use the softened logit of pre-trained teacher network that has higher entropy as an extra supervision to train a student network. KD trains a compact student network to learn not only by the conventional CE loss subjected to the labeled data but also by the final outputs of the teacher network. While KD only utilizes the logit, method such as FitNet (Romero et al., 2014), AT (Zagoruyko & Komodakis, 2016a), FT (Kim et al., 2018) and KTAN (Liu et al., 2018) use the intermediate feature representation to transfer the knowledge of a teacher network.

**Online Knowledge Distillation:** Conventional offline methods require training a teacher model in advance while online methods do not require any pre-trained model. Instead, the networks teach each other mutually by sharing their knowledge throughout the training process. Some examples of recent online methods are DML (Zhang et al., 2018) and ONE (Lan et al., 2018) which demonstrate promising results. DML simply applies KD losses mutually, treating each other as teachers, and it achieves results that is even better than the offline KD method. The drawback of DML is that it lacks an appropriate teacher role, hence provides only limited information to each network. ONE pointed out this defect of DML. Rather than mutually distilling between the networks, ONE generates a gated ensemble logit of the training networks and uses it as a target to align for each network. ONE tries to create a powerful teacher logit that can provide more generalized information. The flaw of ONE is that it can not train different network architectures at the same time due to its architecture of sharing the low-level layers for the gating module. The common limitation of existing online methods is that they are dependent only on the logit and do not make any use of the feature map information. Considering that KD loss term is only applicable to the classification task, transferring knowledge at feature map-level can enlarge the applicability to other tasks. Therefore, our method proposes a distillation method that utilizes not only the logit but also the feature map via adversarial training, moreover, our method can be applied in case where the co-trained networks have different architectures.

**Generative Adversarial Network (GAN):** GAN (Goodfellow et al., 2014) is a generative model framework that is proposed with an adversarial training scheme, using a generator network $G$ and a discriminator network $D$. $G$ learns to generate the real data distribution while $D$ is trained to distinguish the real samples of the dataset from the fake results generated by $G$. The goal of $G$ is to trick $D$ to make a mistake of determining the fake results as the real samples. Though it was initially proposed for generative models, its adversarial training scheme is not limited to data generation. Adversarial training has been adapted to various tasks such as image translation (Isola et al., 2017; Zhu et al., 2017), captioning (Dai et al., 2017), semi-supervised learning (Miyato et al., 2016; Springenberg, 2015), reinforcement learning (Pfau & Vinyals, 2016), and many others. In this paper, we utilize GAN's adversarial training strategy to transfer the knowledge at feature map-level in an online manner. The networks learn the other networks' feature map distributions by trying to deceive the discriminators while the discriminators are trained to distinguish the different distributions of each network.

## 3 PROPOSED METHOD

In this section, we describe the overall process of our proposed *Online Adversarial Feature map Distillation* (AFD). As can be seen in Figure 2, when training two different networks, $\Theta_1$ and $\Theta_2$, in an online manner, we employ two discriminators, $D_1$ and $D_2$. We train $D_1$ such that the feature map of $\Theta_2$ is regarded as a real and that of $\Theta_1$ is classified as a fake and do vice versa for discriminator $D_2$. Then, each network $\Theta_1$ and $\Theta_2$ are trained to fool its corresponding discriminator so that it can generate a feature map that mimics the other network's feature map. Throughout this adversarial training, each network learns the feature map distribution of the other network. By exploiting both logit-based distillation loss and feature map-based adversarial loss together, we could observe

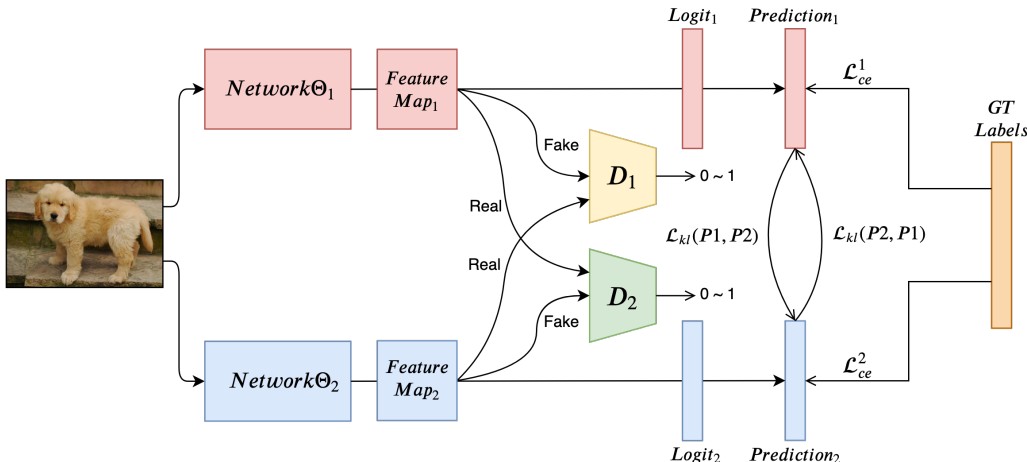

Figure 2: Overall schematic of online adversarial feature map distillation (AFD). At feature map-level, each network is trained to deceive the corresponding discriminator so that it can mimic the other network's feature map distribution. While at logit-level, KL loss to learn the peer network's logit is applied as well as the conventional CE loss.

a significant improvement of performance in various pairs of network architectures especially when training small and large networks together. Also we introduce a cyclic learning scheme for training more than two networks simultaneously. It reduces the number of required discriminators from $2 \times_2 C_K$ (when employing discriminators bidirectionally between every network pairs.) to $K$ where $K$ is the number of networks participating. This cyclic learning framework not only requires less computation than the bidirectional way but also achieves better results compared to other online training schemes for multiple networks.

First, we explain the conventional mutual knowledge distillation method conducted among the networks at the logit-level. Then we introduce our novel online feature map distillation method using the adversarial training scheme in addition to the cyclic learning framework for training more than two networks at the same time.

### 3.1 LOGIT-BASED MUTUAL KNOWLEDGE DISTILLATION

We use two loss terms for logit-based learning, one is the conventional cross-entropy(CE) loss and the other is mutual distillation loss between networks based on Kullback Leibler(KL) divergence. We formulate our proposed method assuming training two networks. Training scheme for more than two networks will be explained in Sec 3.3. Below is the overall logit-based loss for two networks:

$$\mathcal{L}_{logit}^1 = \mathcal{L}_{ce}(y, \sigma(z_1)) + T^2 \times \mathcal{L}_{kl}(\sigma(z_2/T), \sigma(z_1/T)) \tag{1}$$

$$\mathcal{L}_{logit}^2 = \mathcal{L}_{ce}(y, \sigma(z_2)) + T^2 \times \mathcal{L}_{kl}(\sigma(z_1/T), \sigma(z_2/T)). \tag{2}$$

Here, $\sigma(\cdot)$ refers to softmax function and $z \in \mathbb{R}^C$ is the logit produced from a network for $C$-class classification problem. The temperature term $T$ is used to control the level of smoothness in probabilities. As the temperature term $T$ goes up, it creates a more softened probability distribution. We use $T = 3$ for every experiment. $\mathcal{L}_{ce}$ is the CE loss between the ground truth label $y$ and the softmax output $\sigma(z)$ that is commonly used in image classification. $\mathcal{L}_{kl}$ is the KL loss between the softened logit of each network. We multiply the KL loss term with $T^2$ because the gradients produced by the soft targets are scaled by $1/T^2$. While the CE loss is between the correct labels and the outputs of the model, the KL loss is the KL distance between the outputs of two training networks. The KL loss provides an extra information from the peer network so that the network can improve its generalization performance. The difference with DML is that while DML updates asynchronously which means that it updates one network first and then the other network, our AFD updates the networks synchronously, not alternatingly. The CE loss trains the networks to predict the correct truth label while the mutual distillation loss tries to match the outputs of the peer-networks, enabling the networks to share the knowledge at logit-level.

### 3.2 Feature map-based learning via Adversarial Training

Our AFD uses adversarial training to transfer knowledge at feature map-level. We formulate our adversarial feature map distillation for two networks which will be extended for more networks later. We divide a network into two parts, one is the feature extractor part that generates a feature map and the other is the classifier part that transforms the feature map into a logit. Each network also has a corresponding discriminator which distinguishes different feature map distributions. The architecture of the discriminator is simply a series of Conv-Batch_Normalization-Leaky_ReLU-Conv-Sigmoid. It takes a feature map of the last layer and it reduces the spatial size and the number of channel of the feature map as it goes through the convolution operation so that it can produce a single scalar value. Then we apply the sigmoid function of the value to normalize it between 0 and 1.

We utilize the feature extractor part to enable feature map-level distillation. For the convenience of mathematical notation, we name the feature extractor part as $G_k$ and its discriminator as $D_k$, $k$ indicates the network number. As depicted in Figure 2, each network has to fool its discriminator to mimic the peer network's feature map and the discriminator has to discriminate from which network the feature map is originated. Following LSGAN (Mao et al., 2017), our overall adversarial loss for discriminator and the feature extractor can be written as below:

$$\mathcal{L}_{D_1} = [1 - D_1(G_2(x))]^2 + [D_1(G_1(x))]^2 \tag{3}$$

$$\mathcal{L}_{G_1} = [1 - D_1(G_1(x))]^2. \tag{4}$$

The feature extractors $G_1$ and $G_2$ take input $x$ and generate feature maps. The discriminator $D_1$ takes a feature map and yields a scalar between 0 (fake) and 1 (real). It is trained to output 1 if the feature map came from the co-trained network (in this case, $G_2$) or 0 if the feature map is produced from the network it belongs to ($G_1$ in this case). The goal of $D_1$ is to minimize the discriminator loss term $\mathcal{L}_{D1}$ by correctly distinguishing the two different feature map distributions while $G_1$'s goal is to minimize the loss term $\mathcal{L}_{G_1}$ by fooling $D_1$ to make mistake of determining $G_1$'s feature map as real and yield 1. Each training network's object is to minimize $\mathcal{L}_{G_k}$ to mimic the peer network's feature map distribution. This adversarial scheme works exactly the same by changing the role of two networks.

In case where the two networks' feature map outputs have different channel sizes, for example a pair like (WRN-16-2, WRN-16-4) (Zagoruyko & Komodakis, 2016b), we use a transfer layer that is composed of a convolution layer, a batch normalization and a ReLU which converts the number of channels to that of peer network. The above loss terms change as $\mathcal{L}_{D_1} = [1 - D_1(T_2(G_2(x)))]^2 + [D_1(T_1(G_1(x)))]^2$ and $\mathcal{L}_{G_1} = [1 - D_1(T_1(G_1(x)))]^2$ when using the transfer layer $T_k$.

**Optimization:** Combining both logit-based loss and the adversarial feature map-based loss, the overall loss for each network $\Theta_1$ and $\Theta_2$ are as follows:

$$\mathcal{L}_{\Theta_1} = \mathcal{L}_{logit}^1 + \mathcal{L}_{G_1}, \qquad \mathcal{L}_{\Theta_2} = \mathcal{L}_{logit}^2 + \mathcal{L}_{G_2} \tag{5}$$

However, the logit-based loss term $\mathcal{L}_{logit}^k$ and the feature map-based loss term $\mathcal{L}_{G_k}$ are not optimized by the same optimizer. In fact, they are optimized alternatingly in a same mini-batch. At every mini-batch iteration, we infer an image into a model and it computes a logit and a feature map. Then we calculate the two loss terms and optimize the networks based on the two losses separately, meaning that we update the parameters by the logit-based loss once and then update again by the feature map-based loss. The reason we optimize separately for each loss term is because they use different learning rates. The adversarial loss requires much slower learning rate thus if we use the same optimizer with the same learning rate, the networks would not be optimized. Note that we do not infer for each loss term, inference is conducted only once, only the optimization is conducted twice, one for each loss term.

### 3.3 Cyclic Learning Framework

Our method proposes a novel cyclic peer-learning scheme for training more than two networks simultaneously. As can be seen in Figure 3, each network transfers its knowledge to its next peer network in an one-way cyclic manner. If we train $K$ number of networks together, each network distills its knowledge to its next network except the last network transfers its knowledge to the first

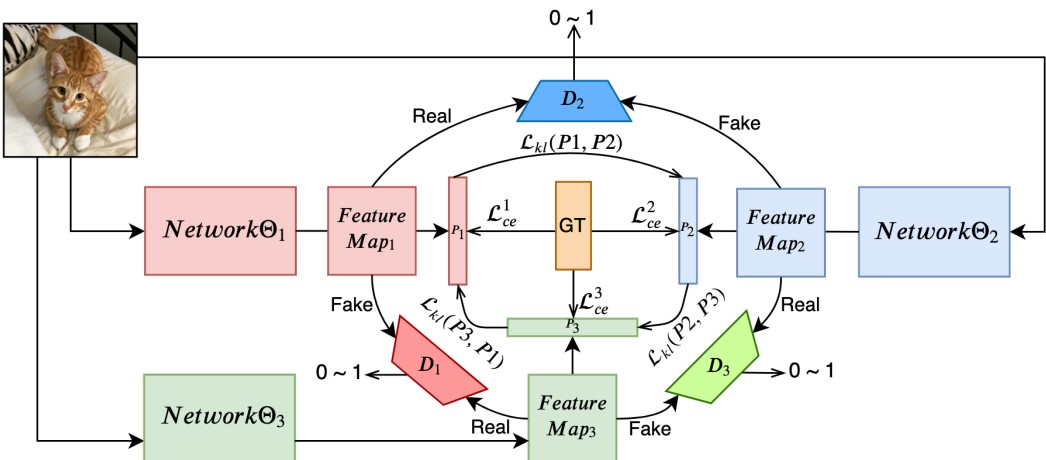

Figure 3: Schematic of cyclic-learning framework for training 3 networks simultaneously.

network, creating a cyclic knowledge transfer flow as $1 \rightarrow 2, 2 \rightarrow 3, \cdots, (K-1) \rightarrow K, K \rightarrow 1$. The main contribution of using this cyclic learning framework is to avoid employing too many number of discriminators. If we apply our adversarial loss for every pair of networks, it would demand two times the amount of every possible pair of $K$ networks which would cost a lot of computation. Also in Sec 4.5, we empirically show that our cyclic training scheme is better than other online methods' training scheme for multiple networks.

## 4 EXPERIMENT

In this section, to show the adequacy of our method, we first present comparison experiment with distance method and ablation study to analyze our method. Then we compare our approach with existing online knowledge distillation methods under different settings. First of all, we demonstrate results on using the same sub-network architectures in Sec 4.3. Then, we apply our method on sub-networks with different architectures in Sec 4.4. In Sec 4.5, we also show the results of training more than two networks to demonstrate that our method generalizes well even when the number of networks increases.

In most of the experiments, we use the CIFAR-100 (Krizhevsky et al.) dataset. It consists of 50K training images and 10K test images over 100 classes, accordingly it has 600 images per each class. All the reported results on CIFAR-100 are average of 5 experiments. Since our method uses two loss terms, logit-based loss and feature map-based loss, we use different learning details for each loss term. For overall learning schedule, we follow the learning schedule of ONE(Lan et al., 2018) to conduct fair comparison which is 300 epochs of training. In terms of logit-based loss, the learning rate starts at 0.1 and is multiplied by 0.1 at 150, 225 epoch. We optimize the logit-based loss using SGD with mini-batch size of 128, momentum 0.9 and weight decay of 1e-4. This learning details for logit-based loss is equally applied to other compared online distillation methods. For feature map-based loss, the learning rate starts at 2e-5 for both discriminators and feature extractors and is decayed by 0.1 at 75, 150 epoch. The feature map-based loss is optimized by ADAM(Kingma & Ba, 2014) with the same mini-batch size and weight decay of 1e-1.

In tables, '2 Net Avg' and 'Ens' represents the average accuracy of the two sub-networks and the ensemble accuracy respectively. The average ensemble is used for AFD, DML and KD while ONE uses gated ensemble of sub-networks according to its methodology.

### 4.1 COMPARISON WITH DIRECT FEATURE MAP ALIGNMENT METHODS

Since our goal is to distill feature map information that suits for mutual online distillation, we briefly compare our method with conventional direct alignment method in Table 1. We train two networks together, in one setting, we use the same architecture (ResNet-32 (He et al., 2016)) and in the other, we use different types (WRN-16-2, WRN-28-2 (Zagoruyko & Komodakis, 2016b)). For $L_1$, each

Table 1: Top-1 accuracy(%) comparison with direct alignment methods using CIFAR-100 dataset.

| Model Type | $L_1$ | | | $L_1+$ KD | | | $L_1+$ KD (offline) | | | AFD | | | Vanilla | |
|---|---|---|---|---|---|---|---|---|---|---|---|---|---|---|
| Same Arch. | 2 Net Avg | | Ens | 2 Net Avg | | Ens | Student | Teacher | Ens | 2 Net Avg | | Ens | Net | |
| ResNet-32 | 66.82 | | 70.69 | 70.16 | | 72.44 | 71.91 | 69.79 | 72.07 | **74.03** | | **75.64** | 69.38 | |
| Different Arch. | Net1 | Net2 | Ens | Net1 | Net2 | Ens | Student | Teacher | Ens | Net1 | Net2 | Ens | Net1 | Net2 |
| WRN-(16-2,28-2) | 69.84 | 73.41 | 74.63 | 72.35 | 74.82 | 75.10 | 73.94 | 73.62 | 76.56 | **75.88** | **77.08** | **77.82** | 71.07 | 73.50 |

Table 2: Ablation study of AFD. Top-1 accuracy(%) on CIFAR-100 dataset.

| Model Type | w/o KD (Adv only) | | | w/o Adv (KD only) | | | Full model (AFD) | | |
|---|---|---|---|---|---|---|---|---|---|
| Same Arch. | 2 Net Avg | | Ens | 2 Net Avg | | Ens | 2 Net Avg | | Ens |
| ResNet-32 | 70.09 | | 74.77 | 73.38 | | 75.21 | 74.03 | | 75.64 |
| WRN-16-2 | 71.94 | | 75.92 | 74.81 | | 76.20 | 75.33 | | 76.34 |
| Different Arch. | Net1 | Net2 | Ens | Net1 | Net2 | Ens | Net1 | Net2 | Ens |
| WRN-(16-2,28-2) | 72.05 | 73.80 | 76.82 | 74.99 | 76.64 | 77.28 | 75.88 | 77.08 | 77.82 |

network is trained not only to follow the ground-truth label by CE loss, but also to mimic the other network's feature map using the $L_1$ distance loss. For $L_1+$ KD, KD (Hinton et al., 2015) loss is applied mutually along with the $L_1$ loss between the feature maps. We also compare our results with offline method, $L1+$ KD (offline) employs a pre-trained network as a teacher network and distills its feature map knowledge to an untrained student network by $L1$ loss as well as the KD loss at logit level. ResNet-32 and WRN-28-2 that shows 69.79% and 73.62% accuracy are used as the teacher networks in the two settings respectively. The results clearly show that learning the distributions of feature maps with adversarial loss performs better than direct alignment method in both mutual online distillation and offline distillation. We could observe that using $L_1$ distance loss actually disturbs the networks to learn good features in online environment. The accuracy of ResNet-32 has dropped more than 2% compared to its vanilla version accuracy (69.38%) and the accuracy of WRN-16-2 is also lower than its vanilla network (71.07%). Even when combined with KD loss($L1$ + KD), direct alignment method shows poor performance compared to ours in both online and offline manner. Though distance loss is used in many conventional offline methods, they suffer when it comes to online environment. In case of different architecture types, our method also outperforms the direct alignment method. It indicates that when it comes to online feature map distillation, transferring feature map information with direct alignment method such as $L1$ distance is worse than indirect distillation that uses feature map distribution via adversarial loss.

## 4.2 ABLATION STUDY

Table 2 shows the ablation study of our proposed method. We conduct experiments using the same and different sub-network architectures. We run three experiments with different training settings for each model case. The three settings are full model, without mutual knowledge distillation at logit-level and without adversarial feature map distillation. When trained without the adversarial feature map distillation, the accuracy decreases in all three model cases. The accuracy of both ResNet-32 and WRN-16-2 dropped by 0.65% and 0.52% respectively, and those of (WRN-16-2, WRN-28-2) pair declined by 0.89% and 0.44% compared to the full model. Ensemble results are also lower than those of the full models. When only the adversarial feature map distillation is applied, the accuracy has increased by 0.71% and 0.87% compared to the vanilla versions of ResNet-32 and WRN-16-2 respectively. Especially in case of different sub-network architecture, the accuracy of WRN-16-2 has increased by almost 1%. Based on these experiments, we could confirm that adversarial feature map distillation has some efficacy of improving the performance in online environment.

## 4.3 SAME ARCHITECTURE

We compare our method with DML and ONE for training two sub-networks with the same architecture. The vanilla network refers to the original network trained without any distillation method. As shown in Table 3, in both ResNet and WRN serises, DML, ONE and AFD all improves the networks' accuracy compared to the vanilla networks. However, AFD shows the highest improvement of performance in both sub-network and ensemble accuracy among the compared distillation methods. Especially in case of ResNet-20, ResNet-32 and WRN-16-2, our method significantly improves the

Table 3: Top-1 accuracy(%) comparison with other online distillation methods for training two same architecture networks as a pair on the CIFAR-100 dataset. The numbers in parentheses refer to the amount of increase in accuracy compared to the vanilla network.

| Model Type | DML | | ONE | | AFD | | Vanila |
|---|---|---|---|---|---|---|---|
| | 2 Net Avg | Ens | 2 Net Avg | Ens | 2 Net Avg | Ens | |
| ResNet-20 | 70.90(+3.42%) | 72.08 | 70.56(+3.08%) | 72.26 | **71.72(+4.24%)** | **72.98** | 67.48 |
| ResNet-32 | 73.40(+4.02%) | 74.89 | 72.61(+3.23%) | 74.07 | **74.03(+4.65%)** | **75.64** | 69.38 |
| ResNet-56 | 75.48(+1.64%) | 76.73 | 76.45(+2.61%) | 77.16 | **77.25(+3.41%)** | **78.35** | 73.84 |
| WRN-16-2 | 74.68(+3.61%) | 75.81 | 73.85(+2.78%) | 74.84 | **75.33(+4.26%)** | **76.34** | 71.07 |
| WRN-16-4 | 78.17(+2.79%) | 79.06 | 77.32(+1.94%) | 77.79 | **78.55(+3.17%)** | **79.28** | 75.38 |
| WRN-28-2 | 77.02(+3.52%) | 78.64 | 76.67(+3.17%) | 77.40 | **77.22(+3.72%)** | **78.72** | 73.50 |
| WRN-28-4 | 79.16(+2.56%) | 80.56 | 79.25(+2.65%) | 79.73 | **79.46(+2.86%)** | **80.65** | 76.60 |

Table 4: Top-1 accuracy(%) comparison with other online distillation methods for training two different architectures as a pair on CIFAR-100 dataset.

| Model Types | | KD | | | DML | | | AFD | | |
|---|---|---|---|---|---|---|---|---|---|---|
| Net1 | Net2 | Net1 | Net2 | Ens | Net1 | Net2 | Ens | Net1 | Net2 | Ens |
| ResNet-32 | ResNet-56 | 72.92 | 76.27 | 76.71 | 73.48 | 76.35 | 76.74 | **74.13** | **76.69** | **77.11** |
| ResNet-32 | WRN-16-4 | 72.67 | 77.26 | 76.94 | 73.48 | 77.43 | 77.01 | **74.43** | **77.82** | **77.67** |
| ResNet-56 | WRN-28-4 | 75.48 | 78.91 | 79.23 | 76.03 | **79.32** | 79.38 | **77.95** | 79.21 | **80.01** |
| ResNet-20 | WRN-28-10 | 70.08 | **78.17** | 76.12 | 71.03 | 77.70 | 75.78 | **72.62** | 77.83 | **76.70** |
| WRN-16-2 | WRN-16-4 | 74.87 | 77.42 | 77.30 | 74.87 | 77.17 | 76.96 | **75.81** | **78.00** | **77.84** |
| WRN-16-2 | WRN-28-2 | 74.86 | 76.45 | 77.29 | 75.11 | 76.91 | 77.24 | **75.88** | **77.08** | **77.82** |
| WRN-16-2 | WRN-28-4 | 74.51 | 78.18 | 77.60 | 74.95 | 78.23 | 77.67 | **76.23** | **78.26** | **78.28** |
| Average | | 73.63 | 77.52 | 77.31 | 74.14 | 77.59 | 77.25 | **75.29** | **77.84** | **77.92** |

accuracy by more than 4% compared to the vanilla version while other distillation methods improve around 3% on average except the ResNet-32 of DML.

## 4.4 DIFFERENT ARCHITECTURE

In this section, we compare our method with DML and KD using different network architectures. We set Net2 as the higher capacity network. For KD, we use the ensemble of the two sub-networks as a teacher to mimic at every iteration. The difference with original KD (Hinton et al., 2015) is that it is an online learning method, not offline. We did not include ONE because ONE can not be applied in case where the sub-networks have different model types due to its architecture of sharing the low-level layers. In table 4, we could observe that our method shows better performance improvement than other methods in both Net1 and Net2 except for a couple of cases. The interesting result is that when AFD is applied, the performance of Net1 (smaller network) is improved significantly compared to other online distillation methods. This is because AFD can transfer the higher capacity network's meaningful knowledge (feature map distribution) to the lower capacity one better than other online methods. When compared with KD and DML, AFD's Net1 accuracy is higher by 1.66% and 1.15% and the ensemble accuracy is better by 0.61% and 0.67% on average respectively. In case of (WRN-16-2, WRN-28-4) pair, the Net1's parameter size (0.70M) is more than 8 times smaller than Net2 (5.87M). Despite the large size difference, our method improves both networks' accuracy, particularly our Net1 performance is better than KD and DML by 1.72% and 1.28% respectively. The performance of KD and DML seems to decline as the difference between the two model sizes gets larger. Throughout this experiment, we have shown that our method also works properly for different architectures of sub-networks even when two networks have large difference in their model sizes. Using our method, smaller network considerably benefits from the large network.

## 4.5 EXPANSION TO 3 NETWORKS

To show our method's expandability for training more than two networks, we conduct experiment of training 3 networks in this section. As proposed in Sec 3.3, our method uses a cyclic learning framework rather than employing adversarial loss between every network pairs in order to reduce

Table 5: Top-1 accuracy(%) comparison with other online distillation methods using 3 networks on CIFAR-100 dataset. '3 Net Avg' represents the average accuracy of the 3 networks.

| Model Type | DML | | ONE | | AFD | | Vanilla |
|---|---|---|---|---|---|---|---|
| | 3 Net Avg | Ens | 3 Net Avg | Ens | 3 Net Avg | Ens | |
| ResNet-32 | 73.43 | 76.11 | 73.25 | 74.94 | **74.14** | **76.64** | 69.38 |
| ResNet-56 | 76.11 | 77.83 | 76.49 | 77.38 | **77.37** | **79.18** | 73.84 |
| WRN-16-2 | 75.15 | 76.93 | 73.87 | 75.26 | **75.65** | **77.54** | 71.07 |
| WRN-28-2 | 77.12 | 79.41 | 76.66 | 77.53 | **77.20** | **79.78** | 73.50 |

Table 6: Top-1 accuracy(%) comparison with DML on ImageNet dataset.

| Model Types | | DML | | | AFD | | | Vanilla | |
|---|---|---|---|---|---|---|---|---|---|
| Net1 | Net2 | Net1 | Net2 | Ens | Net1 | Net2 | Ens | Net1 | Net2 |
| ResNet-18 | ResNet-34 | 70.19 | 73.57 | 73.33 | 70.39 | 74.00 | 74.47 | 69.76 | 73.27 |

the amount of computation and memory. DML calculates the mutual knowledge distillation loss between every network pairs and uses the average of the losses. ONE generates a gated ensemble of the sub-networks and transfers the knowledge of the ensemble logit to each network. As it can be seen in Table 5, AFD outperforms the compared online distillation methods on both 3 Net average and ensemble accuracy in every model types. Comparing the results of Table 5 to that of Table 3, the overall tendency of performance gains compared to DML and ONE is maintained.

## 4.6 IMAGENET EXPERIMENT

We evaluate our method on ImageNet dataset to show that our method can also be applicable to a large scale image dataset. We use ImageNet LSVRC 2015 (Russakovsky et al., 2015) which has 1.2M training images and 50K validation images over 1,000 classes. We compare our method with DML using two pre-trained networks ResNet-18 and ResNet-34 as a pair. The results are after 30 epochs of training. As shown in Table 6, our method improves the networks better than DML.

## 5 CONCLUSION

We proposed an online knowledge distillation method that transfers the knowledge not only at logit-level but also at feature map-level using the adversarial training scheme. Unlike existing online distillation methods, our method utilizes the feature map information and showed that knowledge transfer at feature map-level is possible even in an online environment. Through extensive experiments, we demonstrated the adequacy of adopting the distribution learning via adversarial training for online feature map distillation and could achieve better performance than existing online methods. We also introduced a novel cyclic learning framework for training multiple networks concurrently and presented its efficacy by comparing with existing approaches. We also confirmed that our method is broadly suitable to various architecture types from a very small network (ResNet-20) to a large (WRN-28-4) network. We hope that due to the work of our research, the area of knowledge distillation can be further advanced and studied by many researchers.

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
