# OpenReview forum: "Feature-map-level Online Adversarial Knowledge Distillation"
_ICLR.cc/2020/Conference — Reject_

### Official Review · AnonReviewer1 · 2019-10-17
**Official Blind Review #1**

**Rating:** 3

**Review:**

In this paper, the authors study the online knowledge distillation problem and propose a method called AFD (Online Adversarial Feature map Distillation), which aims to transfers the knowledge of intermediate feature map (first propose) using adversarial training. Then, a cyclic learning scheme is proposed to train more than two networks simultaneously and efficiently. Ablation study on CIFAR100 shows that the adversarial training in AFD can improve the accuracy significantly, while the direct method such as using L1 distance is worse. The comparison experiments with several online distillation methods also show the effectiveness of proposed method.

Some comments or suggestions:
(i) The theoretical analysis is lacking. For example, some formulas proofs can be added to illustrate that the adversarial feature map distillation is more advantageous than the direct feature map alignment.
(ii) The details of the experiments such as parameter configurations are missing, which makes the results not easy to be reproduced.
(iii) Tab.1 and Tab.2 can be combined.


**Experience Assessment:**

I have published one or two papers in this area.

**Review Assessment: Checking Correctness Of Derivations And Theory:**

N/A

**Review Assessment: Checking Correctness Of Experiments:**

I assessed the sensibility of the experiments.

**Review Assessment: Thoroughness In Paper Reading:**

I read the paper at least twice and used my best judgement in assessing the paper.

---

> ### Author Response · Authors · 2019-11-10
> **Author's reply to official blind review #1**
>
> First of all, thank you for all your great comments and reviews. We really appreciate it.
> We did our best to answer your comments, hope it resolves some of questions you have.
>
> 1.It would be great if we can provide some mathematical formulas that can support our idea, but it is very difficult to describe why something is working in knowledge distillation mathematically. However throughout experiments, we could find that direct alignment method only tries to copy each other’s feature map at every iteration and does not really transfer the actual knowledge in online manner. Especially in the early stage of training, directly copying the feature map is not really a good idea since both networks have not yet fully learned the features of the given data. Thus, direct feature map alignment method kills the possibility to learn various features and force both training networks to converge to a point that only minimizes the distance between the feature map of each network. However our method does not fall into these kind of problem because our method is not about directly aligning the feature maps but about transferring the distribution of feature maps. Our method respect each network’s own way to extract features from the data but at the same time it transfers the knowledge of distribution of the learned feature map of each network which can be meaningful.
>
> 2.Implementation details is written in Appendix. If you need more information regarding implementation please ask again at any time.
> Edit : We moved the implementation details that was originally in the appendix section of the first version to the experiment section of the revised version.
>
> 3.We deliberately separated two experiments in order to clarify the purpose of each experiment. Table 1 is to show that direct aligning method using distance loss is not preferable way to transfer the knowledge of feature map in online mutual distillation and insist that our method is more suitable. Table 2 was to analyze the each component of our method.

---

### Official Review · AnonReviewer2 · 2019-10-24
**Official Blind Review #2**

**Rating:** 3

**Review:**

A new online knowledge distillation is investigated by utilizing feature map information next to the logits via GAN. Instead of direct feature map alignment, the algorithm tries to transfer the distribution of the feature maps. There is no teacher per se, but the big and small nets are trained via an adversarial game where 2 discriminators try to minimize the distributions of the two nets. The idea is understandable but some issues remain:
1-	Training GAN is by itself an expensive task and optimization is difficult, so how computationally expensive is this online KD compared to the offline one?
2-	It is not clear form the paper feature maps from which layers are being used? If multiple layers are considered, how did you choose which ones are better. Also the smaller model has a different structure, h ow did you choose to pair feature maps in the big model and the small model?
3-	What would be the performance difference compared to offline knowledge distillation? For example in Table 1 can you please add a column with offline KD?
4-	Mutual training brings some generalization, and when you compare the results in Table 3 with vanilla model, I am wondering if you made sure this is not only due to a better generalization.
5-	In cyclic learning framework, it is not well-motivated why someone wants to train multiple networks that mimic each others’ behavior; also the complexity increases in that case which makes me wondering wouldn’t it be better to do it offline then?


**Experience Assessment:**

I have read many papers in this area.

**Review Assessment: Checking Correctness Of Derivations And Theory:**

I assessed the sensibility of the derivations and theory.

**Review Assessment: Checking Correctness Of Experiments:**

I assessed the sensibility of the experiments.

**Review Assessment: Thoroughness In Paper Reading:**

I made a quick assessment of this paper.

---

> ### Author Response · Authors · 2019-11-10
> **Author's reply to official blind review #2**
>
> First of all, thank you for all your great comments and reviews. We really appreciate it.
> We did our best to answer your comments, hope it resolves some of questions you have.
>
> 1. Well, it is quite hard to compare the computational cost fairly between the online and offline distillation method since their mechanism is quite different. Offline distillation is two phase distillation method which requires pre-training the teacher network in advance, while online distillation method is training multiple networks simultaneously by making them to teach each other. It does not require any pre-trained network. In the sense that offline method requires powerfully pre-trained network, it might need more time and computation. Also in DML paper, it showed that online method presents better performance than offline method. We also have compared the offline method with our method under the same training setting and using the same architecture, and have found out that our method performs better.
>
> 2.Actually these are all written in the paper. In section 1, second to the last paragraph, we have written that we use the feature map from the last convolution layer since deeper convolution layer generates more meaningful features with a high-level abstraction. Also for distilling between different architecture networks, in section 3.2, we have written that we use a transfer layer that is composed of a convolution, a batch normalization and a ReLU to convert the size of peer network to its own network’s feature map size.
>
> 3.Yes we have actually conducted comparison experiment with offline KD but did not include in the paper because we thought comparing offline method and online method is somewhat different problem and a little out of our paper’s focus. However since we already have the experiment results, we would like to share the results here and also revise the paper including the experiment.
>
> In case of the same architecture(res32,res32), we employed a pre-trained res32 network that shows 69.78% accuracy as a teacher network and distill its last layer feature map using L1 loss to another res32 network (student network) from scratch. Also we used the original KD loss at logit level. Thus it is a L1+KD distillation in offline manner.
>
> After training, the student network showed 71.91% accuracy. It is somewhat higher than vanilla accuracy(69.38%) but much lower than ours (74.03%). The results is average of 5 runs of experiment and all the training setting is the same over all experiments.
>
> In case of different architecture network, we used wrn28-2 with 73.62% accuracy as the teacher network and trained wrn16-2 as a student network from scratch. The student network improved upto 73.94% accuracy which is higher than its vanilla accuracy(71.07%) but much lower than ours(75.88%).
>
> 4.For this question, I did not fully understand your question. Mutual distillation improves performance because it enables better generalization capacity of the models. Thus the performance improvement is due to the better generalization of mutual distillation. In previous work, the online mutual distillation was only conducted at logit-level, however what we did is to utilize the information of feature map as well in online manner using the adversarial loss so that we can better train and better improve the performance of the networks.
>
> 5.Training multiple networks that mimic each other is the key and main idea of online distillation. Rather than using a pre-trained static teacher network, online distillation trains multiple networks simultaneously my mimicking each other. It had been shown in DML paper that online method presents better performance than offline method. Also as previously answered in the third question, we also have seen that our method performs better than offline method under the same condition.
> Our paper’s purpose is to develop this online distillation method further by utilizing not only the logit knowledge but also that of the feature map via transferring the distribution of feature maps through adversarial loss.
> The cyclic learning framework is proposed to show that our method can be generalized even when the number of training networks increases. Also in On-the-fly Ensemble paper, their experiments were done using three networks, hence we had to devise a way to apply our method for 3 networks to conduct fair comparison.

---

### Official Review · AnonReviewer3 · 2019-10-26
**Official Blind Review #3**

**Rating:** 6

**Review:**

= Summary
This paper presents a new deep mutual learning (i.e., online peer-teaching) method based on Knowledge Distillation (KD) in a feature map level. The target task is similar with the original KD in the sense that the a network is taught by another network as well as groundtruth labels, but different with the KD in the sense that the networks are not a (frozen) teacher and a student but teaching each other in an online manner. Most approaches in this relatively new line of research rely on logit-based KD for transferring knowledges between networks, and the paper demonstrates that by an additional feature map level KD the performance can be further improved.


= Decision
The current decision is borderline in my mind, but officially weak accept. Although the proposed method is simple and consists of known ideas, it is designed convincingly and enhances performance practically. Also, I believe the target task itself is worth to be introduced as a next direction of KD. However, the submission is weak in terms of novelty and the manuscript should be polished carefully.


= Comments
1) Weak clarity
- The main motivation and advantages of the deep mutual learning is not well introduced in Section 1. Although the two papers (i.e., ONE and DML) are cited here, it would be much better to explicitly describe the main idea and motivation of peer-teaching, what is the difference between the task and the original KD, and the achievements in the previous work. Without these, readers, including me, may get confused why the online KD is required and why there is no clear teacher-student relationship between networks.
- In a similar context, the motivation of introducing more than two student networks should be given.
- The architecture of the discriminator seems not described even in the appendix.
- The meaning of various arrow types in Figure 1 is not clearly described.

2) Insufficient experiments
It would be better to report the performance of vanilla and (offline) KD in Table 1 to show more clearly that the feature map alignment is useful and that online KD is better than its offline counterpart.

3) Limited novelty and performance improvement
- The main idea is already introduced in previous work on the task and the feature map level KD has been studied widely for various applications, their combination is somewhat new though.
- The performance improvement by the proposed feature map level KD seems marginal as shown in Table 2.
- The performance gap between DML and the proposed model seems also marginal.

**Experience Assessment:**

I have read many papers in this area.

**Review Assessment: Checking Correctness Of Derivations And Theory:**

I carefully checked the derivations and theory.

**Review Assessment: Checking Correctness Of Experiments:**

I assessed the sensibility of the experiments.

**Review Assessment: Thoroughness In Paper Reading:**

I read the paper thoroughly.

---

> ### Author Response · Authors · 2019-11-10
> **Author's reply to official blind review #3**
>
> First of all, thank you for all your great comments and reviews. We really appreciate it.
> We did our best to answer your comments, hope it resolves some of questions you have.
>
> 1. Weak clarity
> a)   Thank you for your great advice, I will reflect your comments and revise my paper based on your comments.
> b)   Initially, we included the method and experiment for more than two student networks to show that our method can be applied and generalize well even when the number of networks increases. Also in On-fly-ensemble paper, all their experiments were conducted using three networks, thus we also had to devise a way to extent to multiple networks for fair comparison.
> c)   The architecture of the discriminator is just a series of convolution operation, batch norm and leaky-relu. To be specific, it is Conv-BN-leakyReLU-Conv-sigmoid. We designed that after every convolution operation the spatial size gets smaller so that it can eventually be reduced to a single scalar value at the end. Then we apply sigmoid function to the value so that it can be normalized between 0 and 1.
> d)   The thin line arrow indicates the evolvement of feature map data point as iteration goes on and the broader arrow indicates the way each method compares the feature maps of each network. Thus in figure 1, direct alignment method, directly compares the each data point of feature map using distance loss, but our adversarial feature map distillation method compares the overall distribution of the feature map data points via discriminators which is empirically shown that it actually helps the model to train better and improve the performance.
>
> 2. Insufficient experiment
> a) Actually we also thought about including the results of offline KD in Table 1, but we rather not to because comparing offline method with online method is somewhat different problem. However, since we have the results, we want to share it here and also include in the revised version of paper along with the vanilla accuracy.
> In case of the same architecture(res32,res32), we employed a pre-trained res32 network that shows 69.78% accuracy as a teacher network and distill its last layer's feature map using L1 loss to another res32 network (student network) from scratch. Also we applied the original KD loss at logit level. Thus it is basically a L1+KD distillation in offline manner.
> After training, the student network showed 71.91% accuracy. It is somewhat higher than vanilla accuracy(69.38%) but much lower than ours (74.03%). The results is an average of 5 runs of experiment and all the training setting is the same over all experiments.
> In case of different architecture network, we used wrn28-2 with 73.62% accuracy as the teacher network and trained wrn16-2 as a student network from scratch. The student network improved up to 73.94% accuracy which is higher than its vanilla accuracy(71.07%) but much lower than ours(75.88%).
>
> 3. Limited novelty and performance improvement
> a)  I admit that feature map level KD has been studied in other distillation methods, but they were conducted in off-line manner and also used the method of directly aligning the outputs of teacher network to student such as L1 or L2 losses. We believe that out method is somewhat novel in the sense that it is the first approach to distill feature-map level information in online-manner and that our method does not directly align the feature map outputs but rather indirectly distill using discriminator, transferring the distribution of the feature maps. Also, as you mentioned, the combination of logit-level KD and feature-map level KD is somewhat new and we proved that it has some efficacy by various experiments.
> b)  The performance improvement might seem marginal in the case of the same architecture but in the case of different architecture, the performance improvement is somewhat meaningful, the wrn16-2 trained with only adversarial loss is almost 1% higher than its vanilla, also when trained only with KD loss its accuracy decreases by almost 0.9%.
> It is not included in the paper but we also conducted experiment with  wrn16-2 & wrn28-4 pair. Only with adversarial loss, wrn16-2 had 72.53% accuracy which is higher than vanilla by 1.46% and when trained only with KD loss(without adversarial loss) it showed 75.26% accuracy which is lower by 0.97% than the full AFD model.
> c)  In some model cases the performance gap seems marginal but in the different architecture experiment, the average performance gap between the DML and AFD is more than 1% for the smaller network. Especially in case of (res56-wrn28-4, res20-wrn28-10, wrn16-2-wrn28-4) pairs, the performance gap of the smaller network between DML and ours is more than one percent. Through this experiment our method seems to work well even though the parameter size difference between the two networks is very large. In previous methods, the method seem to perform poorly as the parameter size difference between the two networks become large.

---

> > ### Author Response · Authors · 2019-11-13
> > **Additional comment to 3-c)**
> >
> > It seems that KD and DML seems to suffer when there is a large size difference in the two networks that are trained together. If you compare the (wrn-16-2,wrn-28-2) pair with (wrn-16-2,wrn-28-4) pair for KD and DML, even tough the co-trained network's width has increased, the performance of wrn-16-2 has decreased. The same outcome can be found even when the depth of the co-trained network is increased (wrn-16-2,wrn-16-4) ->(wrn-16-2,wrn-28-4). KD seems to decline the performance of wrn-16-2 and DML increase the performance of wrn-16-2 very marginally. However our method, AFD, evidently improves the smaller network (wrn-16-2) in both cases which the depth or the width of the co-trained network has increased.
> > ((wrn-16-2,wrn-16-4) ->(wrn-16-2,wrn-28-4) : 75.81->76.23,
> >  (wrn-16-2,wrn-28-2) ->(wrn-16-2,wrn-28-4) : 75.88->76.23).
> >
> > We believe that our method has some lesson and is meaningful that it is a first approach to transfer feature map knowledge in online way and discovered the effectiveness of transferring the distribution information of feature map in online manner.

---

### Decision · Program_Chairs · 2019-12-19

**Decision:**

Reject

**Comment:**

The paper received scores of WR (R1) WR (R2) WA (R3), although R3 stated that they were borderline. The main issues were (i) lack of novelty and (ii) insufficient experiments. The AC has closely look at the reviews/comments/rebuttal and examined the paper. Unfortunately, the AC feels that with no-one strongly advocating for acceptance, the paper cannot be accepted at this time. The authors should use the feedback from reviewers to improve their paper.